# Influence of Mechanical Deformations on the Characteristic Impedance of Sewed Textile Signal Lines

**DOI:** 10.3390/ma15031149

**Published:** 2022-02-02

**Authors:** Paweł Kubiak, Jacek Leśnikowski

**Affiliations:** 1Institute of Security Technologies “MORATEX”, Marii Sklodowskiej-Curie 3 Street, 90-505 Lodz, Poland; 2Faculty of Material Technologies and Textile Design, Institute of Architecture of Textiles, Lodz University of Technology, Żeromskiego 116, 90-543 Lodz, Poland; jacek.lesnikowski@p.lodz.pl

**Keywords:** textile signal lines, textronic systems, data transmission over textiles, smart textiles, characteristic impedance, signal quality tests, smart garments, e-textiles

## Abstract

The following article describes a new type of textile signal line that can be used in smart clothing. The article presents the structure of this line and the materials used for its construction. The article also presents the results of research on the influence of the line tensile force on the value of its characteristic impedance. The above tests were carried out on lines where the electrically conductive paths do not have the form of straight lines, as is often the case in smart clothing. The article also presents a preliminary statistical analysis, the aim of which was to find those characteristics of the substrate of the line that affect changes in the characteristic impedance during stretching.

## 1. Introduction

The contemporary textile industry creates many new, unprecedented fields of functionality and opportunities for defined users, by combining traditional textile elements with other products from the field of sciences which are at first glance unrelated to textiles. By combining elements of textiles with electronics and information technology, it is possible to design and manufacture materials with innovative properties and functionalities. The combination of knowledge from these three areas was defined at the Lodz University of Technology in 2003 as a new discipline of knowledge called textronics [1].

Innovative materials, such as intelligent textiles or functional textronic systems are increasingly used in military, specialized or medical technologies. These products are also used in everyday, casual items, such as sportswear, or textiles with worn electronics.

Nowadays, there is a growing demand for innovative textiles with possibilities and applications far different from traditional ones. One of the main reasons for this is the increasing consumer’s awareness, who want modern and fashionable textile products with unusual functions [2]. In consequence, research is focused on innovative materials and solutions increasing the functionality of textile products. Another field of research interest now is the integrated textronic systems combining traditional textiles with electronic elements. These elements allow the processing and transmission of data by electrical signals, their acquisition and management. Also referred to as smart textiles or e-textiles, these systems have a wide range of applications. They can be used, for example, to produce clothing with human physiological parameters monitoring systems [3,4,5,6]. Such clothing can be used in monitoring people working in hazardous conditions [7,8,9], the elderly [10,11] or the chronically ill [12]. Also, textronic clothing used for monitoring the vital parameters of newborns [13,14] is being developed. Textronic applications can also be used in functional clothing for sports [15,16] or clothing for casual users.

Electrically conductive textile materials are used in textronic systems most often to create lines connecting individual electronic systems and elements. These lines can supply low-value electrical energy to these systems or serve as lines to exchange information between them. Lines of this type should have the lowest possible resistance.

The majority of textronic solutions require data exchange between electronic devices of the system. For this purpose, until recently, mainly conventional connections such as wires have been used, but these are elements affecting the ergonomics of the product (stiffness, troublesome maintenance of a textile product with increased weight of the entire product). This resulted in intensive work on an alternative, textile solution. These works concern both the provision of electrical conductivity to textiles and the construction of textile signal lines. The electrical conductivity of textile materials can be achieved in many ways, described, among others, in [17,18].

Such attempts were also made for applying electroconductive layers on the surface of the product (printing, sputtering, embroidering, etc.). The possibility of transmitting electrical signals using electrically conductive textiles is also investigated and developed. The transmission of electrical signals, and in particular high-frequency signals or digital signals, can be a challenge for this type of line. In the presented article, the subject of signal transmission by textile signal lines (TSL) was discussed.

Wired transmission, despite the necessity to use cables between electronic systems, has significant advantages. One of them is simplicity resulting from the lack of necessity to use radio transmitting and receiving systems which usually require an additional power supply and the simplification of electronic circuits and components used. Wired data is also more resistant to naturally occurring signal interference and is also more resistant to eavesdropping. This is important for example in the case of textronic clothing for human physiological parameters monitoring. The advantages of wired data transmission in textronic mean that the construction of such a line made entirely of textile materials is one of the other fields of development. These types of lines should be capable of transmitting signals with a wide frequency spectrum. In addition, they should be resistant to temperature, humidity, mechanical stresses occurring during their use in e-textiles. So far, few published studies of the resistance of textile signal lines to the above factors occurring during the use of e-textiles have been carried out. Leśnikowski [19] investigated the influence of temperature and humidity on textile signal lines in the form of electrically conductive textile strips sewn onto non-conductive fabric. Leśnikowski and Kubiak [20] studied the changes in the characteristic impedance of selected types of textile signal lines during mechanical loads. Leśnikowski [21] also investigated the influence of bending and abrasion of lines on their transmission properties. In all of the above studies, textile signal lines with straight electrically conductive paths were used. In practical applications, textile signal lines must frequently change direction to connect electronic modules placed in different places of the smart garments. Despite this, no results of such studies have been published so far. The article presents studies of the influence of tensile forces acting on the line on its characteristic impedance. As textile signal lines used in smart-garment must change their direction frequently, lines that do not have the form of a straight line were used for the tests. Works related to the subject of data transmission over the surface of textiles also show the importance of adjusting individual elements of the signal path in textronic systems. The appropriate tools, like the Time Domain Reflectometry method [22], for interpreting the quality of the transmitted signal allows the user to identificate potential sources of interference or signal degradation. The article also presents a preliminary statistical analysis, the aim of which was to find those characteristics of the substrate of the line that affect changes in the characteristic impedance during stretching.

## 2. Materials and Methods

For the tests presented in this article, twenty textile signal lines with curved electroconductive paths were made. Each of the lines consists of a non-conductive substrate in the form of a fabric and sewn strips cut from an electroconductive woven fabric. More details on the construction of the lines can be found in [23]. The dimensions of the signal line made by the sewing method were calculated to gain the assumed characteristic impedance equal to 50 Ω. The dimensions were shown in Figure 1, and a physical example of the constructed line is in Figure 2. The shape of the line resulted from the desire to make a line in which the electrically conductive paths would not have the form of a straight line, and at the same time beginning and end of the line would be on one axis. This feature is necessary for the correct tensioning of the line using the method described later in the article.

The paths of the textile signal lines were made of the electro-conductive Soliani Ponge fabric. These paths were sewn to the non-conductive fabrics, creating the substrate of the lines. The Ponge fabric was chosen due to its excellent conductivity as the main factor required for signal transmission. The basic parameters of the Ponge fabric are presented in Table 1, while the basic parameters of the fabrics used as line substrate are presented in Table 2.

In total, twenty different TSLs were made using the same Soliani Ponge electro-conductive material and different material substrates, mainly cotton and polyester to test if any of their substrate parameters may influence the characteristic impedance change. All TSLs are shown in Table 2.

The substrate mass and thickness of the mentioned above, twenty TSL are shown in Figure 3. The lines L1, L2, L3 and L4 were chosen for detailed impedance characteristic waveforms in a further chapter. These lines have a maximum or minimum thickness or surface mass.

All twenty TSLs were divided into groups based on the specified substrate parameters: the thickness, the surface mass, and the substrate material weave. The groups of TSLs according to specified substrate properties were shown in Table 3, Table 4 and Table 5. The plan of dividing the TSLs into groups was to include in each of the three subgroups at least five different lines. Also, the substrates had three specified weaves: plain, twill, and satin, and then all TSL were divided into three groups of the specified weave, the same method used earlier in [20].

The transmission properties of textile signal lines are characterized by different parameters. These parameters determine the line ability to transmit fast-changing and high-frequency signals. One of the crucial parameters is the line characteristic impedance. It is the main value proving that the signal line is matched to other components of the signal path.

The value of the characteristic impedance Z for an ideal (lossless) line is calculated according to the following formula:(1)$$Z=\sqrt{\frac{L}{C}}$$
where L is the line inductance and C is the line capacitance.

In the real signal line there are natural changes in series resistance of the conductive elements used, or dielectric losses [24]. Thus, the impedance of the real lines including naturally present losses is calculated by:(2)$$Z=\sqrt{\frac{R_{S}+j\omega L}{G+j\omega C}\;}$$
where Rs is the unit series resistance of the conducting part of the line, G means unit conductance of the dielectric and ω is the pulsation (rad/s).

When designing and developing transmission systems, their elements should be matched to the system and the impedance of each element should be as close to each other as possible. The value of the characteristic impedance of the line depends on its application, e.g., for lines connecting a radio transmitter with a textile antenna, this value is usually 50 Ω. The suitability of signal lines with a characteristic impedance of 50 Ω for signal transmission was discussed by Johnson [23]. These tests showed that the value 50 Ω of the characteristic impedance is a compromise for the impedance value between 30 Ω, at which the maximum power of the transmitted signal occurs, and the value of 77 Ω, where the signal losses are the lowest.

The value of the TSL characteristic impedance depends mainly on the geometrical dimensions of such elements, the dielectric properties of the substrate and the resistance of the material from which the electrically conductive paths were made. Earlier, only straight TSLs were made and tested [20]. In a straight signal line, the electro-conductive paths are in the form of straight lines. In the studies presented in [20], the change of TSL transmission properties was tested before and after subjecting them to tensile loads. Tensile loads were obtained by loading one end of the line with a mass of a certain weight. In smart clothing, it is very often necessary to use TSL, which are not straight lines. This is due to the need to change the direction of the lines to connect the electronic modules that can be placed in different places of the smart garment. As mentioned earlier such studies have not been carried out so far. The research on these types of lines is presented in the article below.

In the research conducted, the so-called characteristic impedance profile of the tested lines has been measured. This profile shows the value of the characteristic impedance of each point of the line as a function of its distance from the beginning of the line. To determine these profiles, the reflectometric method described, among others in [22,25] was used. The measuring stand for testing the TSL’s characteristic impedance was built with the Tektronix DSA8200 Digital Serial Analyzer, connected with the 80A02 EOS Electric Charge Protection module, with an additional external 80E08 TDR/Sampling Module. The block diagram of the stand is shown in Figure 4. The tested line is connected to the TDR module using the terminals described in detail in [26]. These clamps enable the connection of the flat electro-conductive paths of the tested line with concentric measuring connectors of the measuring equipment. The Digital Serial Analyzer, using the 80E08 module, generates a voltage step that is applied to the input of the tested line. The generated wave propagates along the tested line reflecting from places with different impedance. The reflected signal returns to the analyzer. Based on its changes over time, the line impedance profile is determined.

The characteristic impedance profiles were collected via the Serial X-Press software extension of the DSA8200 Signal Analyzer. All measurements were made in a standardized, normal climate: 20 °C and air humidity 65% according to the ISO standard [27].The impedance profile of each line connected to the measurement system (Figure 4) was measured in real-time. The measured TSL was mounted vertically, with the end unloaded (m_0_ = 0 g) and tested. Then each of the lines was successively loaded with the masses m_1_ = 0.414 kg and m_2_ = 1.461 kg and tested. The example of a line mounted vertically with load m_2_ = 1.461 kg was shown in Figure 5.

These two mass values were experimentally chosen as a simulation of mechanical forces that may occur in real-time deformations in the textiles when wearing. Choosing less mass than proposed results in deformations which are hardly visible, when adding too much mass could lead to permanently deform or even damage constructed TSLs. The reason for vertically stretching the lines was ease of obtaining a reproducible tensile force. For this purpose, test weights with known mass were used.

## 3. Results and Discussion

The measured characteristic impedance profiles of the transmission lines (Figure 6, Figure 7, Figure 8 and Figure 9) are characterized by a certain non-uniformity. This phenomenon has a fundamental and direct impact on the quality of the electric signal transmitted through the line. In the case of an ideal line, the value of the characteristic impedance should be constant at every point on the line, but in reality, there will always be some fluctuation in waveforms. The value of these factors also depends on the quality of the implementation of individual line elements.

The observed characteristic impedance waveform for line L1 (with the lowest surface mass) shows some fluctuations in the range of 60 to 80 Ω and some minor differences when loading with mass m_1_ or m_2_ in comparison to the unloaded line. At the points of curvature of the lines, no significant changes in characteristic impedance under stretching were observed. This means that line bends do not significantly affect the transmission properties of this type of line.

Characteristic impedance waveform for line L2 (with the lowest substrate thickness) shows more significant fluctuations from 60 to 80 Ω and above, at the end of the unloaded line. Also, some major differences, especially when loading with m_2_ mass in comparison to the unloaded line, were observed. Significant changes in the characteristic impedance were observed in the places where the electrically conductive paths were bent. This can mean a significant change in the impedance of the line if it is placed where it will be exposed to stretching.

The L3 line (with the highest surface mass) shows the characteristic impedance waveform very similar to L1, although the overall impedance value is lower by about 10 Ω in comparison to other lines. This line also has some fluctuations from 50 to 70 Ω and some minor differences when loading with mass m_1_ or m_2_ compared to the unloaded line. This line is very similar in its characteristic impedance waveform compared to line L1 and due to the lower values obtained it is the most suitable for applications in textronic systems with the main impedance equal to 50 Ω.

The observed characteristic impedance waveform for line L4 (with the highest substrate thickness) shows only minimal fluctuations from 60 to 85 Ω and some minor differences when loading with mass m_1_ or m_2_ compared to the unloaded line. This means that the line can correctly transmit signals when placed in areas of the garment where stress occurs.

The value of the characteristic impedance of the tested line depends on the dimensions of its electrically conductive paths and the spacing between them. In particular, the distance between the ground paths significantly influences this value. Stretching the lines longitudinally changes the dimensions of the electrically conductive paths and spacing between them. The degree of these changes may depend on many parameters characterizing the structure of the substrate and the electro-conductive paths of the lines. One of the aims of the research presented in the article below was to check which of the parameters characterizing the line substrate have a significant impact on the change of the characteristic impedance of the line under the influence of its stretching. To assess this, the statistical analysis presented later in the article was performed.

## 4. Statistical Analysis

For statistical analysis, the average value from each impedance profile was determined according to the formula:(3)$$Z_{\mathrm{av}}=\frac{\sum_{i=1}^{n}Z_{i}}{n}$$
where Z_av_ is the average line impedance, Zi is the line impedance at the i-th point of the average impedance profile, n is the number of points (measurement values) of which the line impedance profile consists.

The characteristic impedance of each of the twenty tested lines was measured five times in each load condition (unloaded, loaded with mass m_1_, loaded with mass m_2_). Next, these five impedance waveforms for each load were averaged, resulting in one waveform for specified line without load, for load m_1_ and m_2_. For the statistical analysis, there was a need to compare fixed values instead of waveforms. Therefore, each averaged waveform was shortened, to one average value, using Equation (3).

A non-parametric test was used instead of the multifactor analysis of variance (ANOVA). This was due to failure to meet the assumptions of analysis of variance, namely a lack of normality of distributions in groups determined by variables, and a lack of homogeneity of variance.

The analyzed parameters of all the signal lines population (twenty lines in total) were divided into more than two groups, thus the statistical test must be appropriate to compare its parameters within several independent groups. For that, the nonparametric Kruskal–Wallis statistical test was used. The significance level at all tests was assumed to be α = 0.05. A summary of the collected results showing the characteristic impedance values of all twenty made curved TSL under the influence of various loads is given in Figure 10.

In Figure 10 mean and maximum values of characteristic impedance significantly decrease while loading masses m_1_ and m_2_. The influence on such results is directly related to the way the TSL is stretched under m_1_ and m_2_ loads, which significantly alters the mutual arrangement of the electrically conductive strips which are part of the TSL.

Next, the percentage change of characteristic impedance within different groups of TSLs was tested to find if any of the substrate parameters: surface mass, thickness and weaves (Table 3, Table 4 and Table 5) have a statistical impact. The results are shown in Figure 11, Figure 12 and Figure 13. The mean value, mean +/− Standard Deviation and Max/Min values are also shown.

The statistical analysis of the obtained results consisted in assuming the H0 hypothesis assuming the equality of the mean impedance value in the tested groups. Then, the Kruskal–Wallis test was used to verify this hypothesis.

The Kruskal–Wallis statistical test, with its null hypothesis H0, shows the impact on the value of lines characteristic impedance by the mechanical load placed to the line with the weights of m_1_ and m_2_. Also, the impact from substrate properties—surface mass, thickness and weave to affect the characteristic impedance changes—were tested. The assumed null hypothesis H0 assumes no statistically significant differences between the tested groups of cases. As a result of this test, the probability *p*-value is obtained that the assumed hypothesis H0 is true. If the probability value *p* is greater than the assumed significance level (α = 0.05), then the H0 hypothesis should be accepted. Otherwise, it would be rejected. For example, the values of the TSL characteristic impedance with mass m_1_ and m_2_ loads are shown in Figure 10. The figure shows that the mean value of the impedance is lower when the force stretching the lines is higher. This was confirmed by the result of the Kruskal–Wallis test based on which the assumed null hypothesis was rejected (Table 6). It is equivalent to a statistically significant influence of the tested factor on the characteristic impedance of the line.

The highlighted factors (*p* > 0.05) means that hypotheses H0 that states the mechanical load or substrate properties affects the characteristic impedance changes are accepted. The results of the statistical analysis presented in Table 6 show that the load of the line with the mass m_1_ and in particular mass m_2_ has a statistically significant influence on the changes of the characteristic impedance. Therefore, a further statistical analysis was carried out to show which property of the line substrate may have a statistically significant impact on impedance changes under the influence of these loads. The results of this analysis are presented in Table 6 which shows the characteristics that have such an effect are the surface mass and the thickness of the substrate material. The weave of the substrate has no effect. Considering collected results, there is a possibility to make a TSL within the limits of predetermined values, and the main difficulty to gain this is the line accuracy and precision in the production method. Constructing other TSLs with different shapes and predetermined values is also possible. As seen in Figure 6, Figure 7, Figure 8 and Figure 9, the characteristic impedance was around 50–80 Ω. The suitability of lines with such characteristic impedance spread depends on the specific application and data transmission standard.

## 5. Conclusions

Textronic systems based on electrical elements nowadays have an increasing range of applications. Data transmission via signal lines, constructed with the use of textile elements, will allow improving the products ergonomic by limiting the usage of the conventional elements (cables, connectors, etc.).

Statistically, constructed curved textile signal lines show significant changes in characteristic impedance between unloading state and load with mass m_1_ and in particular mass m_2_. Curved TSL were made strictly to sewn to textiles and can be placed even in custom shape textiles carried by users. The changes of their shapes don’t affect their characteristic impedance, and observed changes were seen when loading with significant mass only. Also, no significant impedance changes were observed at the curvature points of the line in comparison to the straight, non-curved areas of lines. In most cases, no curvature impact for the characteristic impedance change was observed for lines loaded up to mass m_2_. Statistically for the sewed TSLs, only the substrate weave does not influence the characteristic impedance change, regardless of the tensile forces. The surface weight and thickness of the fabric from which the line substrate is made has a significant effect on the changes in the characteristic impedance under the influence of stretching. This impact is less for lines with a thick substrate or substrate having a high surface mass. This agrees with our subjective feeling.

The tensile sensitivity of TSL means that these lines in smart clothing should be led and applicable through places that are not very exposed to mechanical deformation. If the line has to run through such places, it should be placed on a thick substrate made of fabric with a large surface mass.

## Figures and Tables

**Figure 1 materials-15-01149-f001:**
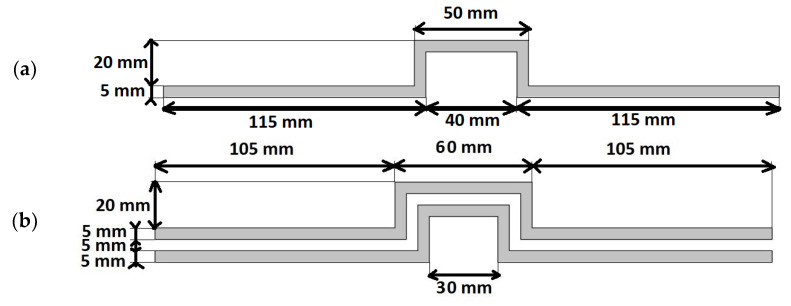
Geometric dimensions of the textile signal line. View from the side of the signal path (**a**), View from the side of the ground paths (**b**).

**Figure 2 materials-15-01149-f002:**
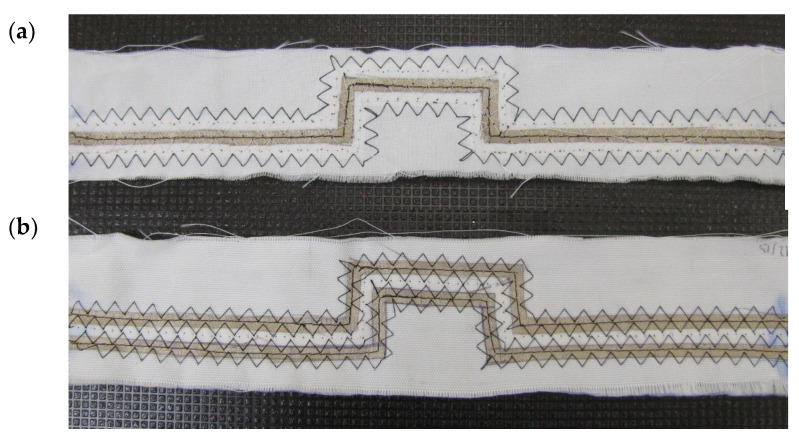
Signal lines produced by sewing technique. View from the side of the signal path (**a**), view from the side of the ground paths (**b**).

**Figure 3 materials-15-01149-f003:**
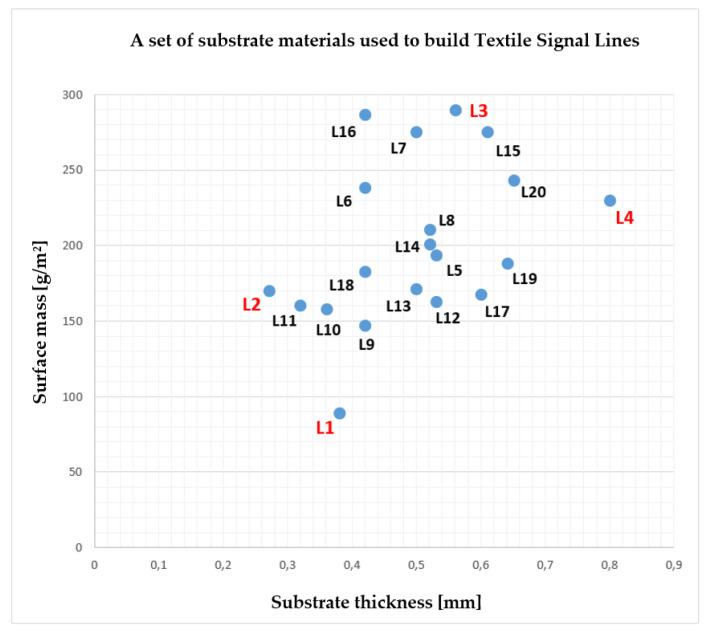
Chart with twenty sewed textile signal lines with their substrate properties.

**Figure 4 materials-15-01149-f004:**
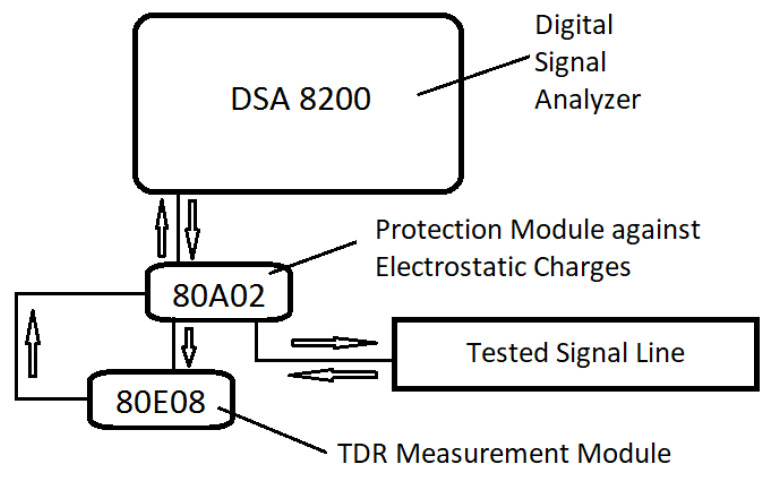
Block diagram of the measuring method for the TSL’s characteristic impedance.

**Figure 5 materials-15-01149-f005:**
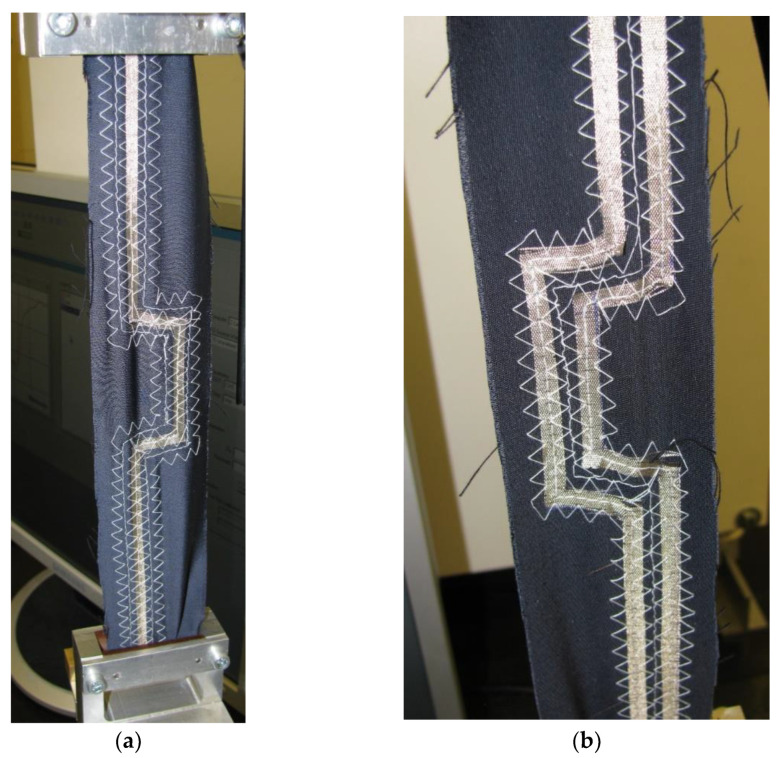
Textile signal line mounted vertically with load m_2_ = 1.461 kg placed to the end of the line. View from the side of the signal path (**a**), View from the side of the ground paths (**b**).

**Figure 6 materials-15-01149-f006:**
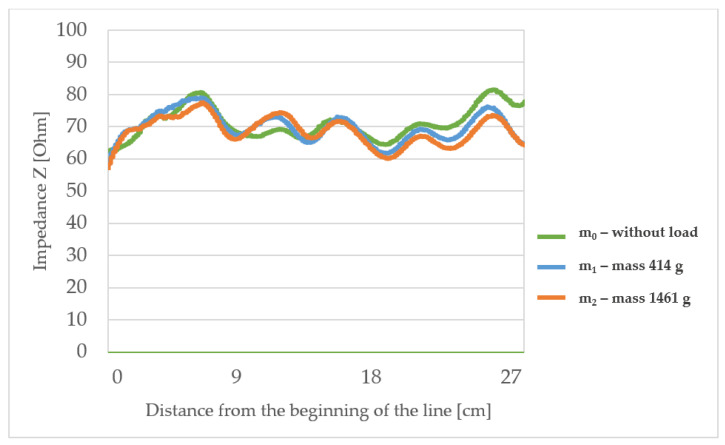
The profiles of the characteristic impedance of the textile signal line L1.

**Figure 7 materials-15-01149-f007:**
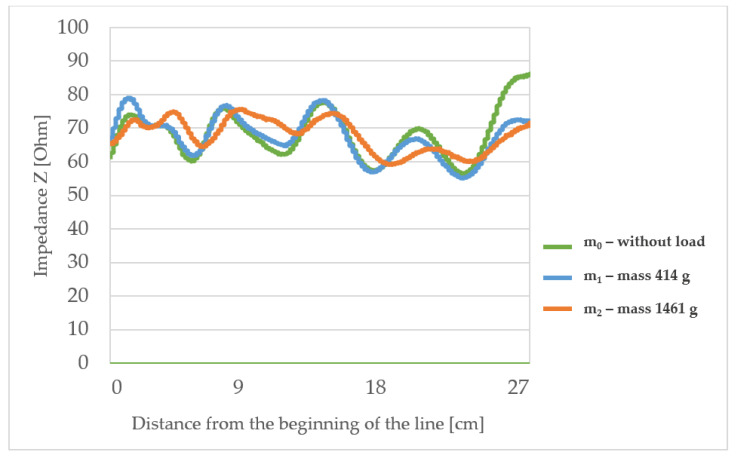
The profiles of the characteristic impedance of the textile signal line L2.

**Figure 8 materials-15-01149-f008:**
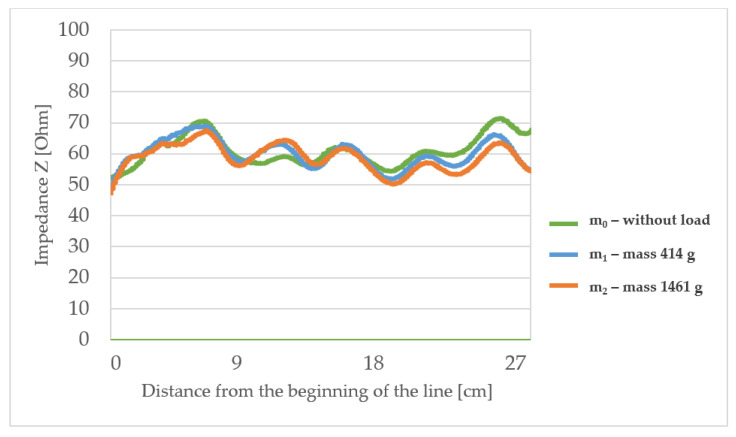
The profiles of the characteristic impedance of the textile signal line L3.

**Figure 9 materials-15-01149-f009:**
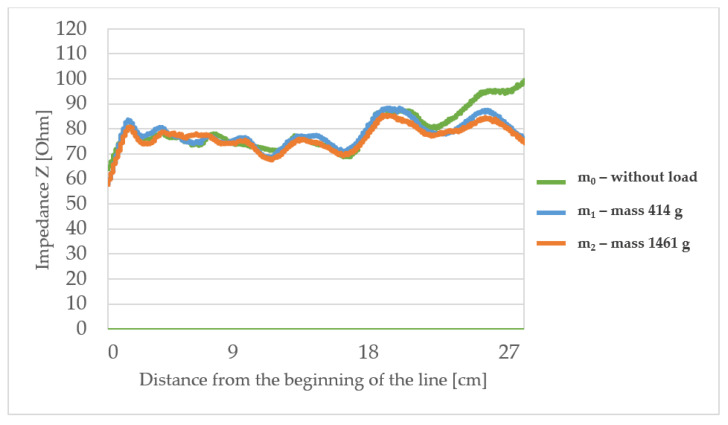
The profiles of the characteristic impedance of the textile signal line L4.

**Figure 10 materials-15-01149-f010:**
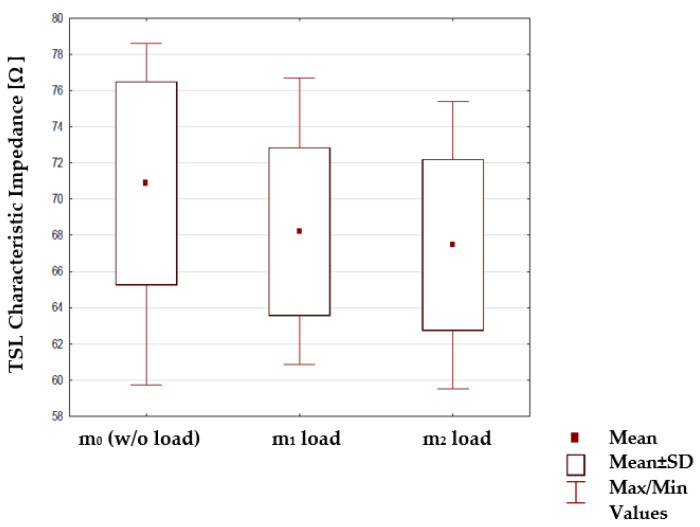
The characteristic impedance of lines under different loads for all sewed lines.

**Figure 11 materials-15-01149-f011:**
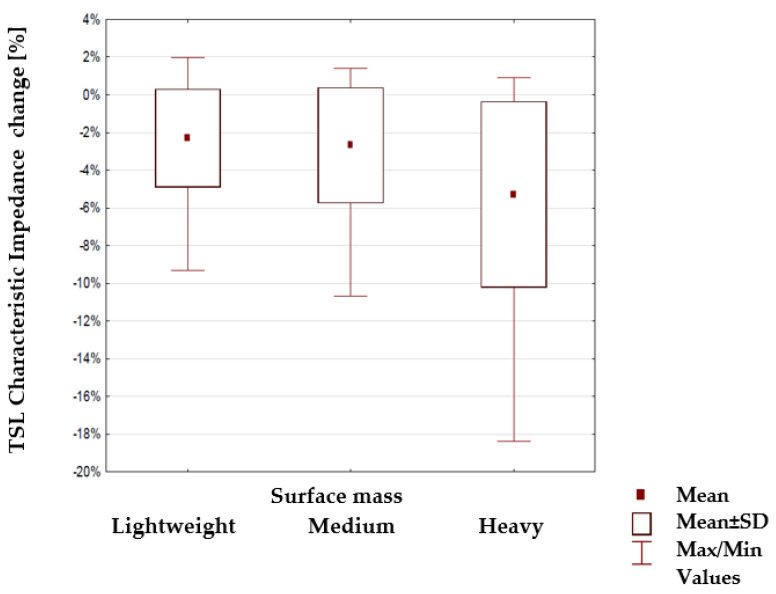
Changes in the line characteristic impedance under load for lines with a different surface mass of the substrate.

**Figure 12 materials-15-01149-f012:**
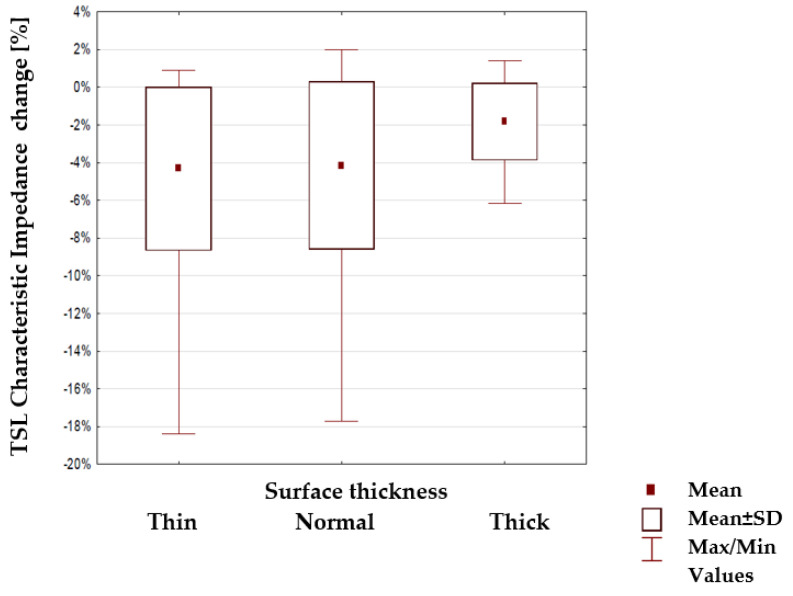
Changes in the line characteristic impedance under load for lines with a different thickness of the substrate.

**Figure 13 materials-15-01149-f013:**
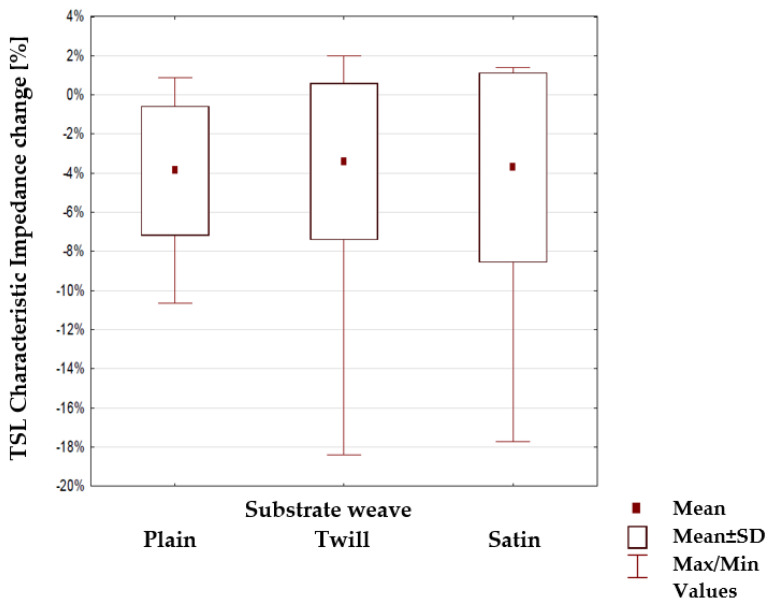
Changes in the line characteristic impedance under load for lines with a different weave of the substrate.

**Table 1 materials-15-01149-t001:** The basic parameters of Ponge fabric are used as electro-conductive paths.

Material	Trade Name/Producer	Thickness (mm)	Surface Resistivity (Ohm/sq)	Nickel Amount (g/m^2^)	Total Weight (g/m^2^)	Weave	Warp Density (Yarns/dm)	Weft Density (Yarns/dm)
Nickel metallised polyester	Ponge/Soliani	0.15	Max. average 0.4	16	60 ± 15	Plain	260	180

**Table 2 materials-15-01149-t002:** The basic parameters of fabrics are used as a non-electroconductive substrate.

Line No.	Material	Thickness (mm)	Surface Mass (g/m^2^)	Weave	Warp Density (Yarns/dm)	Weft Density (Yarns/dm)
L1	100% Cotton	0.38	89	Plain	25	21
L2	65% Polyester, 35% Cotton,	0.27	170	Plain	42	29
L3	65% Polyester, 35% Cotton,	0.56	290	Twill	40	24
L4	40% Polyester, 35% Cotton, 25% Flax	0.80	230	Plain	22	18
L5	69% Cotton, 31% Polyamide	0.53	194	Satin	73	31
L6	63% Polyester, 33% Cotton, 1% Elastomer	0.42	238	Satin	112	25
L7	60% Polyester, 40% Cotton,	0.50	275	Satin	50	28
L8	100% Polyamide	0.52	211	Satin	42	24
L9	100% viscoze	0.42	147	Plain	45	44
L10	100% Polyester	0.36	158	Plain	56	28
L11	100% Polyamide	0.32	161	Plain	25	22
L12	72% Cotton, 23% Polyester, 5% Elastomer	0.53	163	Plain	48	28
L13	55% Flax, 45% Viscose	0.50	172	Plain	24	14
L14	100% Viscose	0.52	201	Plain	22	18
L15	62% Polyester, 32% Viscoze, 6% Elastomer	0.61	275	Plain	33	31
L16	100% Wool	0.42	287	Plain	24	20
L17	63% Cotton, 37% Polyamide	0.60	168	Twill	59	45
L18	100% Polyester,	0.42	183	Twill	36	34
L19	50% Polyester, 50% Wool	0.64	188	Twill	21	15
L20	50% Polyester, 50% Argon (Viscoze-based)	0.65	243	Twill	27	25

**Table 3 materials-15-01149-t003:** TSL’s population, divided into groups based on their substrate thickness.

Substrate Thickness [mm]	Group Name (Thickness)	Number of Lines in the Group
0.27–0.45	Thin	5
0.45–0.59	Normal	8
0.60–0.80	Thick	7

**Table 4 materials-15-01149-t004:** TSL’s population, divided into groups based on their substrate surface mass.

Surface Mass [g/m^2^]	Group Name (Surface Mass)	Number of Lines in the Group
89–165	Lightweight	8
166–229	Medium	7
230–290	Heavy	5

**Table 5 materials-15-01149-t005:** TSLs population, divided into groups based on their substrate weave.

Weave	Number of Lines in the Group
Plain	11
Twill	5
Satin	4

**Table 6 materials-15-01149-t006:** Kruskal-Wallis non-parametrical test summary for curved TSL.

	Characteristic Impedance *Z* [Ohm] Change between Mass Loaded	
Factor or substrate property tested	Between loads with mass m_0_ and m_1_	Between loads with mass m_0_ and m_2_
Mass loading	*p* = 0.003	*p* = 0.001
Surface mass of the substrate	*p* = 0.0022	*p* = 0.001
Substrate thickness	*p* = 0.013	*p* = 0.001
Substrate weave	***p* = 0.1176**	***p* = 0.1721**

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
