# Peer review of "Influence of Mechanical Deformations on the Characteristic Impedance of Sewed Textile Signal Lines"

_materials, 2022, doi:10.3390/ma15031149_

Round 1
Reviewer 1 Report
The function and stability of the power and data transmitting lines is of great importance in textile-based devices. However, the lines in textiles are faced with complex environment during daily use. I think the authors choosed a very practical research point which is really critical for textronics. For the publication, I would like to give some suggestions.
- The authors introduced the prospects, principles and applications of the power and data transmitting lines in great detail. A series of factors, such as temperature, humidity, and deformation, that may affect the performance of the wire are put forward, but in the follow-up research, only the characteristic resistance curve under different weaves and different degrees of deformation is studied. Whether there will be a feeling of top-heavy, it is recommended to focus on the key factors of verification.
- The author gave the characteristic resistance changes under different waves and loads, but did not explain what structural changes caused the signal fluctuations. Could you please further elaborate on some structure-related mechanisms and put forward some suggestions for improvement.
- Can the author provide some higher-quality images? In some graphs, there are more overlapping curves, which can easily lose data details.
- The authors should carefully cheak the typos of the manuscript.
Author Response
The author would like to thank the Associate Editor and the Reviewers for giving their valuable time in carefully reading the manuscript and providing constructive comments and suggestions to improve the quality of the manuscript.
The author has revised the paper accordingly. All modifications and changes are indicated by highlighting the text.
The following responses have been prepared to address all of the Reviewers’ remarks.
Please find the responses attached.

Reviewer 2 Report
Dear Authors,
Kindly go through my comments attached here

Author Response

(The authors gave the same response as above.)

Reviewer 3 Report
- the author didn’t organize the manuscript logically and they spent too much illustration on what others have done rather than their own work.
- No details of the methodology and materials they used could be found in the manuscript. Besides, error ranges of the data are too broad that the statistics are not representative. I suggest expanding the sample size and doing more statistical analysis.
Author Response

(The authors gave the same response as above.)

Round 2
Reviewer 2 Report
Well answered